# Simultaneous Computation and Memory Efficient Zeroth-Order Optimizer for Fine-Tuning Large Language Models

## Abstract

Fine-tuning is powerful for adapting large language models to downstream tasks, but it often results in huge memory usages. A promising approach to mitigate this is using Zeroth-Order (ZO) optimization, which estimates gradients to replace First-Order (FO) gradient calculations, albeit with longer training time due to its stochastic nature. By revisiting the Memory-efficient ZO (MeZO) optimizer, we discover that the full-parameter perturbation and updating processes consume over 50% of its overall fine-tuning time cost. Based on these observations, we introduce a novel layer-wise sparse computation and memory efficient ZO optimizer, named LeZO. LeZO treats layers as fundamental units for sparsification and dynamically perturbs different parameter subsets in each step to achieve full-parameter fine-tuning. LeZO incorporates layer-wise parameter sparsity in the process of simultaneous perturbation stochastic approximation (SPSA) and ZO stochastic gradient descent (ZO-SGD). It achieves accelerated computation during perturbation and updating processes without additional memory overhead. We conduct extensive experiments with the OPT model family on the SuperGLUE benchmark and two generative tasks. The experiments show that LeZO accelerates training without compromising the performance of ZO optimization. Specifically, it achieves over $3\times$ speedup compared to MeZO on the SST-2, BoolQ, and Copa tasks.

## 1 Introduction

Large language models (LLMs) have shown remarkable capabilities in understanding and generating languages, and have been widely adopted in various applications (Brown et al., 2020; Wei et al., 2022; Rao et al., 2022; Wang et al., 2023). To effectively adapt LLMs to downstream tasks, full-parameter fine-tuning has become crucial (Li & Liang, 2021; Lester et al., 2021; Hu et al., 2022; Malladi et al., 2023). However, as the scale of LLMs continues to increase, the memory usage and computational cost of fine-tuning also escalate (Kaplan et al., 2020; Hoffmann et al., 2022), posing a significant challenge to the practical application of LLMs.

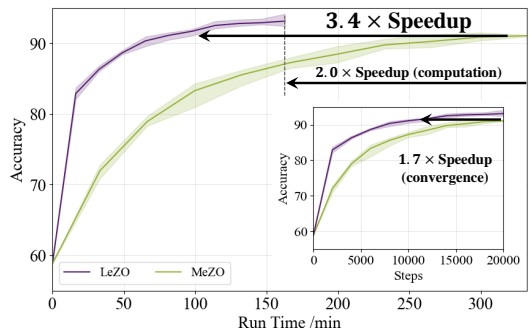

Figure 1: LeZO achieves a $3.4\times$ speedup for fine-tuning the OPT-13b model in run-time compared with MeZO on the SST-2 dataset.

To address these challenges, researchers have proposed efficient fine-tuning methods, which can be categorized into forward-backward (Li & Liang, 2021; Hu et al., 2022; Ding et al., 2022; Pu et al., 2023; Xu et al., 2023) and forward-only (Zhang et al., 2024; Liu et al., 2024; Guo et al., 2024) approaches. Forward-backward methods typically use First-Order (FO) optimizers (Robbins & Monro, 1951; Kingma & Ba, 2015), updating either partial model parameters or introducing small trainable modules to reduce memory usage. However, the performance of these methods can hardly match full-parameter fine-tuning. Dynamic update schemes (Brock et al., 2017; Liu et al., 2021; Pan et al., 2024) address this problem by dynam-

icly changing learnable parameter subsets, but they still require caching many activations. In contrast, forward-only methods utilize Zeroth-Order (ZO) optimizers, which estimate gradients without back-propagation, thereby reducing the memory overhead associated with forward-backward methods. Malladi et al. (2023) first introduce a memory-efficient ZO optimizer (MeZO) for LLM fine-tuning. ZO optimizers, however, exhibit higher stochasticity in gradient estimation compared to FO optimizers, and this stochasticity increases as the scale of tuned parameters grows (Wang et al., 2018; Balasubramanian & Ghadimi, 2018; Cai et al., 2022). Consequently, MeZO requires significantly more computational cost than FO optimizers.

In this paper, we revisit MeZO and identify that its computational cost is primarily in the forward pass, perturbation, and updating stages. Specifically, perturbation and updating account for over 50% of the total time when fine-tuning the OPT-13B model (Zhang et al., 2022) on the SST-2 task. Simplifying these steps can accelerate MeZO. One promising approach is to sparsify the tuned parameters, which has been shown to be effective for FO optimization (Pan et al., 2024).

Inspired by these insights, we propose a layer-wise sparse and efficient zeroth-order optimizer (LeZO). LeZO aims to improve computational efficiency without additional memory overhead. We use layers as the basic unit of sparsity, dynamically updating the set of layers to be tuned at different fine-tuning steps. We integrate this policy into Simultaneous Perturbation Stochastic Approximation (SPSA) and Zeroth-Order Stochastic Gradient Descent (ZO-SGD), which enables sparse ZO optimization through localized parameter perturbation and updating. Over multiple steps, LeZO can achieve full-parameter fine-tuning. LeZO accelerates computations because, during each step, the weights of untuned layers remain dense. This allows us to skip perturbation and updating processes for those layers, thereby reducing floating-point computations. Additionally, compared to MeZO, LeZO does not introduce extra memory overhead. We employ a simple and efficient random selection strategy, avoiding the need for new parameter modules.

We evaluate our approach by fine-tuning models from the OPT family on the SuperGLUE benchmark (Wang et al., 2019), and the SQuAD (Rajpurkar et al., 2016) and DROP (Dua et al., 2019) datasets. By sparsifying 75% of the layers and fine-tuning full-parameters, LeZO outperforms MeZO on most datasets across three model scales. For instance, on the OPT-13B model, LeZO surpasses MeZO by an average of 1% in accuracy or F1 score across seven datasets. When integrating the ZO optimizer with Parameter Efficient Fine Tuning (PEFT), LeZO exceeds MeZO on five datasets, highlighting the versatility and effectiveness of our sparsity scheme. Additionally, LeZO achieves over $2\times$ computational acceleration across some tasks, with speedups increasing as layer sparsity exceeds 75%. Finally, LeZO achieves over $3\times$ training acceleration for the OPT-13B model fine-tuning on the SST-2, BoolQ, and Copa datasets.

In summary, the contributions of this paper are as follows:

- We analyze the computational costs of MeZO during LLM fine-tuning and find that the perturbation and updating processes account for over 50% of the fine-tuning time cost.
- We propose a dynamic layer-wise sparse and efficient zeroth-order optimizer that effectively reduces computational cost in fine-tuning LLMs, and achieve faster convergence than MeZO.
- We evaluate LeZO on SuperGLUE and two generative tasks. This demonstrates its acceleration capability across different sparsity rates and showcases its performance improvement.

## 2 RELATED WORK

**Forward-Backward Fine-Tuning Optimziers** rely on the FO optimizer and compute gradients through derivation (Robbins & Monro, 1951; Kingma & Ba, 2015), which results in high memory usage. One approach to handle this problem is Parameter-Efficient Fine-Tuning (PEFT) (Xu et al., 2023). Relevant strategies involve adding trainable modules (Li & Liang, 2021; Hambardzumyan et al., 2021) or learnable input embeddings (Houlsby et al., 2019) to the model. Moreover, the LoRA series (Hu et al., 2022; Dettmers et al., 2024) adopt low-rank decomposition to reduce the size of trainable modules. However, recent studies show that partial fine-tuning cannot match the performance of full-parameter fine-tuning (Ding et al., 2022; Pu et al., 2023). To handle this problem, some methods enable full-parameter fine-tuning by adjusting parameters in an interleaved fashion. For example, LISA (Pan et al., 2024) updates parameters of different layers in different iterations

based on layer importance. Galore (Zhao et al., 2024) switches low-rank subspaces using the rank information while simultaneously modifying the learnable parameters. These techniques still require storing the intermediate states of FO optimizers, resulting in significant memory overhead.

**Forward-Only Fine-tuning Optimziers** do not require back-propagation for gradient computation (Malladi et al., 2023; Zhang et al., 2024). Malladi et al. (2023) firstly introduce the ZO optimizer for LLMs fine-tuning, known as MeZO, which significantly reduces memory costs. Because MeZO employs SPSA to estimate gradients, it exhibits greater randomness compared to FO optimizers. Consequently, it requires more computational cost to converge. To accelerate the convergence of MeZO, Sparse-MeZO (Liu et al., 2024) only updates model parameters with small values to enhance the effectiveness of ZO optimizer. Nevertheless, this approach necessitates ranking parameter values and introduces a mask matrix; therefore, it results in additional memory and computational overhead. Guo et al. (2024) reduced the number of learnable parameters to one-thousandth. However, it requires cloud computation to provide first-order gradient of the objective function, which is unsuitable for more general scenarios. In summary, how to effectively accelerate the convergence speed of ZO optimizers without incurring additional memory overhead still remains a challenge.

Our proposed LeZO adopts ZO optimization to update model parameters, reducing memory usage compared to forward-backward fine-tuning. By employing a dynamic layer-wise sparsity strategy, LeZO significantly reduces the number of trainable parameters and minimizes the computational costs during the forward-only fine-tuning. Thus, LeZO achieves high efficiency in both memory usage and computational cost.

## 3 EFFICIENCY ANALYSIS OF MEZO

In this section, we first revisit the classical ZO gradient estimator SPSA (Spall, 1992) and the ZO optimizer ZO-SGD (Malladi et al., 2023). Subsequently, we analyze the computational redundancy of MeZO (Malladi et al., 2023).

### 3.1 PRELIMINARIES

SPSA approximates gradients for multi-parameter systems by applying random perturbations, which eliminates the need of gradient back-propagation. It iteratively optimizes parameters by estimating gradients based on differences in function values around a given point. SPSA does not require the function to be smooth or convex, but it generally converges slowly.

**Definition 1** (Simultaneous Perturbation Stochastic Approximation, SPSA (Spall, 1992)). *Given a group of parameters $\boldsymbol{\theta} \in \mathbb{R}^d$ from a model and a loss function $\mathcal{L}$ used for optimization, SPSA estimates the gradient on a mini-batch of data $\mathcal{B}$ as follows:*

$$\widehat{\nabla}\mathcal{L}(\boldsymbol{\theta}; \mathcal{B}) = \frac{\mathcal{L}(\boldsymbol{\theta} + \epsilon \boldsymbol{z}; \mathcal{B}) - \mathcal{L}(\boldsymbol{\theta} - \epsilon \boldsymbol{z}; \mathcal{B})}{2\epsilon} \boldsymbol{z} \approx \boldsymbol{z}\boldsymbol{z}^\top \nabla\mathcal{L}(\boldsymbol{\theta}; \mathcal{B}), \tag{1}$$

*where $\boldsymbol{z} \in \mathbb{R}^d$ with $\boldsymbol{z} \sim \mathcal{N}(0, \boldsymbol{I}_d)$ and $\epsilon$ is a* perturbation scale.

Malladi et al. (2023) applied SPSA (Definition 1) to optimize LLMs, using it to obtain approximate gradients of parameters. They modified the SGD algorithm (Robbins & Monro, 1951), creating ZO-SGD (Definition 2), which incorporates ZO differential gradients to update model parameters. MeZO employs both SPSA and ZO-SGD, and performs two forward passes in a single optimization step without caching the activation information of the parameters. Therefore, it reduces memory usage to the size occupied by the model parameters alone.

**Definition 2** (Zeroth-Order Stochastic Gradient Descent, ZO-SGD (Malladi et al., 2023)). *Given a learning rate $\eta$ and a group of parameters $\boldsymbol{\theta} \in \mathbb{R}^d$, the parameters at time $t$ can be updated using the SPSA gradient estimate as follows:*

$$\boldsymbol{\theta}_{t+1} = \boldsymbol{\theta}_t - \eta\widehat{\nabla}\mathcal{L}(\boldsymbol{\theta}; \mathcal{B}_t), \tag{2}$$

*where $\mathcal{B}_t$ is the mini-batch of data used at time $t$.*

### 3.2 EFFICIENCY DILEMMA OF MEZO

Fine-tuning LLMs with MeZO is extremely time-consuming. It often takes hundreds or even thousands of times longer than using FO optimizers like SGD (Robbins & Monro, 1951) and Adam (Kingma & Ba, 2015). We break down the MeZO computation process into three stages: forward pass, perturbation, and updating. Algorithm 1 explains the computational operations for each of these stages. Figure 2 illustrates their relative time cost in an optimization step.

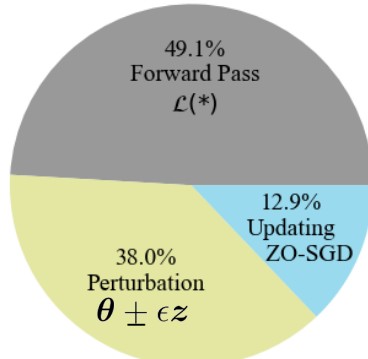

Surprisingly, in classification tasks, the parameter perturbation and updating stages account for more than one half of the overall time. A common method to reduce the computational burden during optimization is to skip the calculations of certain modules (Fan et al., 2019). However, this approach typically introduces an inherent error during the forward pass (Evans & Aamodt, 2021). This phenomenon is challenging to mitigate and usually slows down the convergence rate. An alternative strategy is to selectively optimize the parameters of specific modules during the updating stage. Many studies have substantiated the efficacy of this strategy in FO optimization (Liu

Figure 2: Proportion of computational time cost for each operation in a single step when fine-tuning the OPT-13b model on the SST-2 dataset using MeZO.

et al., 2021; Mi et al., 2022; Pan et al., 2024; Zhao et al., 2024). The above observations motivate us to implement sparse perturbation and updating for MeZO.

## 4 METHODOLOGY

In this section, we present the implementation details of LeZO. We explain how it achieves convergence and computation acceleration without additional memory consumption by updating only a subset of structured parameters during each optimization step.

### 4.1 DYNAMIC LAYER-WISE SPARSE ZEROTH-ORDER OPTIMIZATION

Building upon Equation (1), we apply structured pruning to the model parameter vector $\boldsymbol{\theta}$. In each iteration, we partition this vector into two distinct parts. We introduce a sparse function $\mathcal{R}$ to extract sub-vectors from the input vector and stipulate this partitioning is structured by layer. Given a sparse rate $\rho$ $(0 \leq \rho \leq 1)$, a parameter vector $\boldsymbol{\theta}$, and a random number $s$, we obtain a new parameter vector $\boldsymbol{\theta}' \in \mathbb{R}^{\rho d}$, which is mathematically represented as:

$$\boldsymbol{\theta}' = \mathcal{R}(\boldsymbol{\theta}, \rho, s_t). \tag{3}$$

In each step, we maintain the number of elements in $\boldsymbol{\theta}'$. The random seed $s_t$ is used to determine which layers' parameters are selected to form $\boldsymbol{\theta}'$.

**Definition 3** (Layer-wise Sparse SPSA). *During each step, the sparse parameter set is used as a condition for differential calculation in SPSA, which is defined as follows:*

$$\widehat{\nabla}\mathcal{L}_{\mathcal{R}}(\boldsymbol{\theta}; \mathcal{B}) = \frac{\mathcal{L}(\boldsymbol{\theta} + \epsilon\boldsymbol{z}'; \mathcal{B}) - \mathcal{L}(\boldsymbol{\theta} - \epsilon\boldsymbol{z}'; \mathcal{B})}{2\epsilon}\boldsymbol{z}', \tag{4}$$

*where $\boldsymbol{z}' = \mathcal{R}(\boldsymbol{z}, \rho, s_t)$.*

Furthermore, incorporating layer-wise sparsity into ZO-SGD results in:

**Definition 4** (LeZO-SGD). *At time $t$, $\boldsymbol{\theta}$ can be updated using the layer-wise sparse SPSA gradient estimation as:*

$$\boldsymbol{\theta}'_{t+1} = \boldsymbol{\theta}'_t - \eta\widehat{\nabla}\mathcal{L}_{\mathcal{R}}(\boldsymbol{\theta}; \mathcal{B}_t). \tag{5}$$

Algorithm 1 presents the pseudocode for LeZO, employing a straightforward approach to implement the sparse function $\mathcal{R}$. We use **layers** as the fundamental unit for sparsity (Fan et al., 2020). By maintaining a subset $a$ to store pruned layers, these layers are skipped during perturbation and parameter updating processes.

---

**Algorithm 1:** LeZO (Layer-wise Sparse and Efficient Zeroth-Order Optimization)

---

**Require**: parameters $\boldsymbol{\theta} \in \mathbb{R}^d$, loss $\mathcal{L} : \mathbb{R}^d \to \mathbb{R}$, step budget $T$, perturbation scale $\mu$, batch size $B$, learning rate schedule $\{\eta_t\}$, layer number $N$, and dropping layer number $n \in (0, N)$.

**for** $t = 1, ..., T$ **do**
    Sample batch $\mathcal{B} \subset \mathcal{D}$ and random seed $s$
    Randomly select $n$ elements from $\{1, ..., N\}$ to generate subset $a$
    $\boldsymbol{\theta} \leftarrow$ **PerturbParameters** $(\boldsymbol{\theta}, a, \mu, s)$             ▷ Perturbation
    $\ell_+ \leftarrow \mathcal{L}(\boldsymbol{\theta}; \mathcal{B})$             ▷ Forward Pass
    $\boldsymbol{\theta} \leftarrow$ **PerturbParameters** $(\boldsymbol{\theta}, a, -2\mu, s)$             ▷ Perturbation
    $\ell_- \leftarrow \mathcal{L}(\boldsymbol{\theta}; \mathcal{B})$             ▷ Forward Pass
    $\boldsymbol{\theta} \leftarrow$ **PerturbParameters** $(\boldsymbol{\theta}, a, \mu, s)$             ▷ Perturbation
    `projected_grad` $\leftarrow (\ell_+ - \ell_-)/(2\mu)$
    Reset random number generator with seed $s$
    **for** $\theta_i \in \boldsymbol{\theta}$ & $i \notin a$ **do**
        $z \sim \mathcal{N}(0, 1)$
        $\theta_i \leftarrow \theta_i - \eta_t *$ `projected_grad` $* z$             ▷ Updating
    **end**
**end**

**Subroutine** **PerturbParameters** $(\boldsymbol{\theta}, a, \mu, s)$
    Reset random number generator with seed $s$
    **for** $\theta_i \in \boldsymbol{\theta}$ & $i \notin a$ **do**
        $z \sim \mathcal{N}(0, 1)$
        $\theta_i \leftarrow \theta_i + \mu z$
    **end**
    **return** $\boldsymbol{\theta}$

---

*Remark* 1. As illustrated in Algorithm 1, it is evident that LeZO introduces a layer-wise selection operation $\mathcal{R}$ compared to MeZO. Given the scale of parameters in the billions, this addition has a negligible computational impact. We have the following remarks. **(1)** During the Forward pass, LeZO's computation aligns with MeZO, incurring no additional memory or computational overhead. **(2)** During the perturbation process, we skip the parameter perturbation in the sparsified layers, thereby reducing the computational load. Both LeZO and MeZO utilize a random seed for fixed perturbation $z$. This approach eliminates the need for additional cache storage for perturbation information, thereby not increasing memory overhead. The same principle applies to the updating process. Therefore, it is evident that LeZO reduces computational overhead through the introduction of layer-wise parameter sparsification during the perturbation and updating processes and incurs no additional memory overhead.

### 4.2 CONVERGENCE ANALYSIS OF LeZO

When examining the individual optimization steps, LeZO effectively updates a sub-network, similar to Sparse-MeZO (Liu et al., 2024). We aim to demonstrate that as model parameters converge towards local optimal values, there exists a value $\sigma^2$ such that the speed at which the loss approaches a local optimum is directly proportional to the number of parameters. This implies that the parameter convergence rate is related to $\rho$.

**Lemma 1** (Unbiased Estimation of Sparse Gradient). *Let* $\mathcal{L}_{\mathcal{R}} = \mathbb{E}_{\mathcal{R}}[\mathcal{L}(\boldsymbol{\theta} + \epsilon \boldsymbol{z}')]$. *The relationship between the model's sparse gradient* $\widehat{\nabla}_{\boldsymbol{\theta}'} \mathcal{L}_{\mathcal{R}}(\boldsymbol{\theta})$ *and the estimated ZO sparse gradient* $\widehat{\nabla}\mathcal{L}_{\mathcal{R}}(\boldsymbol{\theta})$ *can be expressed as:*

$$\widehat{\nabla}_{\boldsymbol{\theta}'} \mathcal{L}_{\mathcal{R}}(\boldsymbol{\theta}) = \mathcal{R}(\nabla \mathcal{L}_{\mathcal{R}}(\boldsymbol{\theta})) = \nabla_{\boldsymbol{\theta}'} \mathbb{E}_{\mathcal{R}}[\mathcal{L}(\boldsymbol{\theta} + \epsilon \boldsymbol{z}')]$$
$$= \mathbb{E}_{\mathcal{R}}\left[\frac{\mathcal{L}(\boldsymbol{\theta} + \epsilon \boldsymbol{z}') - \mathcal{L}(\boldsymbol{\theta} - \epsilon \boldsymbol{z}')}{2\epsilon} \boldsymbol{z}'\right] = \mathbb{E}_{\mathcal{R}}[\widehat{\nabla}\mathcal{L}_{\mathcal{R}}(\boldsymbol{\theta})]. \tag{6}$$

From Lemma 1, we can see that the estimated gradient by sparse ZO is an unbiased estimation of the model's sparse parameter gradient.

Table 1: Experiments on the OPT-13b model. SSZO sparsifies 75% of the layers (30 layers out of 40). Two A100-40G GPUs are used for BoolQ and SQuAD tasks.

| Task | SST-2 | RTE | CB | BoolQ | WSC | WIC | Copa | SQuAD | AVG. |
|------|-------|-----|-----|-------|-----|-----|------|-------|------|
| Task type | | | — classification — | | | | multiple choice | generation | |
| Zero-Shot | 58.8 | 59.6 | 46.4 | 59.0 | 38.5 | 55.0 | 80.0 | 46.2 | 55.4 |
| ICL | 87.0 | 62.1 | 57.1 | 66.9 | 39.4 | 50.5 | 87.0 | 75.9 | 65.7 |
| FT (12× memory) | 92.0 | 70.8 | 83.9 | 77.1 | 63.5 | 70.1 | 79.0 | 84.9 | 77.7 |
| MeZO (Malladi et al., 2023) | 91.4 | 66.1 | 67.9 | 67.6 | 63.5 | 61.1 | 88.0 | 84.7 | 73.8 |
| MeZO (Reproduce) | 91.1±0.1 | 70.8±1.4 | 68.8±1.8 | 69.1±0.6 | 63.5±1.2 | 58.7±0.9 | 87.0±1.6 | 84.2±1.0 | 74.2±1.1 |
| SSZO | 93.0±0.3 | 71.4±2.1 | 69.2±1.7 | 74.3±0.8 | 62.7±0.4 | 60.7±1.5 | 87.2±0.8 | 84.3±0.7 | 75.4±1.1 |

Table 2: Experiments on OPT-1.3b model. LeZO sparsifies 75% of the layers (18 layers out of 24).

| Task | SST-2 | RTE | CB | BoolQ | WSC | WIC | MultiRC | Copa | ReCoRD | SQuAD | DROP |
|------|-------|-----|-----|-------|-----|-----|---------|------|--------|-------|------|
| Task type | | | — classfication — | | | | | — multiple choice — | | —- generation —- | |
| zero-shot | 53.6 | 53.4 | 39.3 | 61.1 | 43.3 | 57.5 | 45.4 | 75.0 | 70.6 | 27.2 | 11.2 |
| ICL | 67.7 | 53.1 | 44.6 | 67.2 | 53.8 | 56.0 | 44.7 | 70.0 | 69.9 | 59.0 | 20.3 |
| MeZO | 89.7±1.3 | **65.4±2.5** | 69.3±3.0 | 64.1±0.8 | **63.5±0.0** | 58.7±0.5 | 60.4±1.5 | 76.0±1.9 | 71.6±0.5 | **76.9±0.8** | 23.1±0.3 |
| LeZO | **91.9±0.3** | 64.5±2.4 | **69.6±1.8** | **65.3±1.2** | 63.3±1.1 | **59.4±1.3** | **61.4±1.2** | **76.8±2.3** | **71.7±0.4** | 76.8±0.7 | **24.1±0.8** |

**Assumption 1** (Lipschitz Continuous). *Let $\nabla\mathcal{L}(\boldsymbol{\theta}; \boldsymbol{x})$ denotes the first-order gradient of $\mathcal{L}$ with respect to $\boldsymbol{\theta}$ at $x$. $\mathcal{L}$ satisfies Lipschitz continuity, then*

$$\|\nabla\mathcal{L}(\boldsymbol{\theta}; \boldsymbol{x}) - \nabla\mathcal{L}(\boldsymbol{\theta}_t; \boldsymbol{x})\| \leq \frac{L(l)}{2}\|\boldsymbol{\theta} - \boldsymbol{\theta}_t\|^2, \tag{7}$$

*where $L(l)$ is a constant ensuring $\mathcal{L}$ satisfies Lipschitz continuity.*

Assuming that the loss function $\mathcal{L}(\boldsymbol{\theta}; \boldsymbol{x})$ is Lipschitz continuous (Assumption 1), we can calculate the norm distance between the sparse ZO estimated gradient and the true sparse gradient $\nabla\mathcal{L}_{\mathcal{R}}(\boldsymbol{\theta})$. This distance, denoted as $\|\widehat{\nabla}_{\boldsymbol{\theta}'}\mathcal{L}_{\mathcal{R}}(\boldsymbol{\theta}) - \nabla\mathcal{L}_{\mathcal{R}}(\boldsymbol{\theta})\|$, helps establish their relationship using Lemma 1.

**Lemma 2** (Relationship between Sparse Gradient and Estimate Value). *Let the loss function $\mathcal{L}(\boldsymbol{\theta}; \boldsymbol{x})$ to be Lipschitz continuous. According to Minkowski inequality, we have*

$$\|\nabla\mathcal{L}_{\mathcal{R}}(\boldsymbol{\theta})\|^2 \leq 2\|\widehat{\nabla}_{\boldsymbol{\theta}'}\mathcal{L}_{\mathcal{R}}(\boldsymbol{\theta})\|^2 + \frac{\epsilon^2 L^2(l)}{2}(\rho d + 4)^3, \tag{8}$$

*where $\nabla\mathcal{L}_{\mathcal{R}}(\boldsymbol{\theta}) = \mathcal{R}(\nabla\mathcal{L}(\boldsymbol{\theta}))$.*

Finally, we can obtain the convergence rate of LeZO.

**Lemma 3** (Convergence of LeZO). *Assuming a sequence of generated parameters $\{\boldsymbol{\theta}_t\}_{t\geq 0}$ in LeZO. We have*

$$\mathbb{E}_{\mathcal{R},x}[\|\nabla_{\boldsymbol{\theta}}\mathcal{L}_{\mathcal{R}}(\boldsymbol{\theta}_T)\|^2] \leq \sigma^2 \tag{9}$$

*for any $T = \mathcal{O}(\frac{\rho dL}{\sigma^2})$. $t$ represents the step of fine-tuning and $L(l) \leq L$ for all $\mathcal{L}(\boldsymbol{\theta}_t)$.*

From Lemma 3, it is apparent that as the sparsity ratio $\rho$ decreases, the upper bound on the time required to converge to the expected value also decreases. Hence, parameter sparsity and the smoothness of the objective function can enhance the convergence speed.

## 5 EXPERIMENTS

In this section, we present extensive experiments to validate the effectiveness of the LeZO algorithm.

### 5.1 EXPERIMENTAL SETTING

**Models and Datasets.** To evaluate the efficacy of LeZO, we follow the validation methodology by Malladi et al. (2023). We conduct experiments on the OPT model family (Zhang et al., 2022) at various scales: 1.3 billion, 13 billion, and 30 billion parameters, with 32, 40, and 48 Transformer blocks, respectively. These models utilize the core components of the Transformer architecture (Vaswani et al., 2017), including self-attention mechanisms, feed-forward neural networks,

Table 4: Fine-tuning performance of the ZO optimizer with PEFT. †indicates the results reported in the paper of Malladi et al. (2023). LeZO (LoRA/prefix) outperforms MeZO (LoRA/prefix) on 4 out of 5 tasks. LeZO (LoRA) sparsifies 50% of layers, while LeZO (prefix) sparsifies 75%.

| Task | SST-2 | CB | BoolQ | Copa | SQuAD |
|---|---|---|---|---|---|
| MeZO (LoRA)† | 89.6 | 66.1 | 73.8 | 84.0 | 83.8 |
| MeZO (LoRA) | $90.6_{\pm1.6}$ | $70.4_{\pm1.6}$ | $71.8_{\pm1.2}$ | $86.0_{\pm1.2}$ | $81.1_{\pm0.8}$ |
| MeZO (prefix)† | 90.7 | 69.6 | 73.1 | 87.0 | 84.2 |
| MeZO (prefix) | $90.7_{\pm0.9}$ | $67.9_{\pm2.5}$ | $73.2_{\pm0.4}$ | $87.4_{\pm1.7}$ | $83.3_{\pm1.0}$ |
| LeZO (LoRA) | $\mathbf{92.3_{\pm0.5}}$ | $\mathbf{71.1_{\pm0.8}}$ | $73.7_{\pm0.8}$ | $86.4_{\pm1.5}$ | $81.5_{\pm1.1}$ |
| LeZO (prefix) | $92.1_{\pm0.7}$ | $69.7_{\pm2.5}$ | $\mathbf{74.5_{\pm0.6}}$ | $\mathbf{87.6_{\pm1.1}}$ | $83.1_{\pm0.4}$ |

residual connections, and layer normalization within each block. For downstream tasks, we use the SuperGLUE benchmark (Wang et al., 2019), which includes classification tasks, multiple-choice tasks, and two question-answering tasks.

**Methods.** We utilize zero-shot and 32-shot In-Content Learning (ICL) (Brown et al., 2020) alongside fine-tuning with Adamw (FT) as comparison. To demonstrate LeZO's ability to accelerate model convergence and reduce time cost per training step without additional memory costs, we compared it with MeZO (Malladi et al., 2023). In addition to full-parameter fine-tuning, we integrate LeZO with two PEFT methods: LoRA (Hu et al., 2022) and prefix tuning (Li & Liang, 2021), to further enhance fine-tuning efficiency.

**Hyper-parameters.** We follow most of the experimental settings used in MeZO, including configurations for LoRA and prefix tuning, and methods for selecting training and testing data. We conduct a grid search for the learning rate and perturbation scale to identify the optimal parameters for each experimental group. We use the experimental code for ZO optimization provided by Zhang et al. (2024) and set the validation steps to 2000. Experiments are conducted on an A100-40G GPU. However, due to its memory limitations compare to the A100-80G GPU used by Malladi et al. (2023), we employ a multi-card parallel strategy for fine-tuning the OPT-13b and 30b model on certain downstream tasks to ensure parameter consistency. We select the final model based on the checkpoint with the lowest validation loss. All numerical results report in the experiments are reproduced. We conduct parameter fine-tuning experiments using five different random seeds, reporting the average and standard deviation of these results. Detailed settings are provided in Appendix A. *Unless otherwise specified, all baseline experiments are conducted using the OPT-13b model with 75% sparsity (30 out of 40 layers).*

Table 3: Experiments on the OPT-30b model. LeZO sparsifies 75% of the layers (36 layers out of 48). We report the best results of MeZO and MeZO with prefix from the paper of Malladi et al. (2023). Two A100-40G GPUs are used for SST2, and four for BoolQ.

| Task | SST-2 | BoolQ |
|---|---|---|
| zero-shot | 56.7 | 39.1 |
| ICL | 81.9 | 66.2 |
| MeZO/MeZO (prefix) | 90.6 | 73.5 |
| MeZO (reproduce) | $90.3_{\pm0.5}$ | $69.5_{\pm0.4}$ |
| LeZO | $\mathbf{92.8_{\pm0.6}}$ | $\mathbf{73.7_{\pm0.9}}$ |

## 5.2 MAIN RESULT

**Comparison between LeZO and Other Methods.** LeZO outperforms both non-training methods and MeZO, and surpasses the FO optimizer on some tasks. Table 1 presents a comparison between LeZO and the baseline methods such as zero-shot, the training-free method (ICL), Fine-Tuning (FT) with AdamW for FO optimization, and MeZO across various downstream tasks. We obtain the following observations. **1)** LeZO significantly outperforms non-training methods such as zero-shot and ICL on all tasks. This demonstrates LeZO's superior convergence capabilities during training. **2)** LeZO surpasses MeZO in 7 out of 8 tasks, with an average improvement exceeding 1% across all tasks. Notable enhancements are observed on the SST-2, BoolQ, and WIC datasets, with average accuracy increases over 2% across five experiments. This indicates that layer-wise sparsity schemes can effectively accelerate convergence. However, there is a slight performance degradation on the WSC task compared to MeZO, likely due to the inherent instability of ZO optimization methods and MeZO's sensitivity to prompts. It is speculated that LeZO may inherit these drawbacks, as the fundamental differential gradient computation process in SPSA remains unchanged. **3)** LeZO marginally outperforms FT on the SST-2, RTE, and Copa datasets, while demonstrating comparable

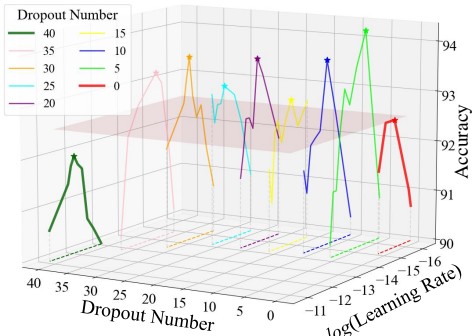
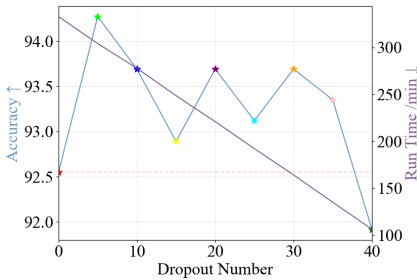

Figure 3: Impact of learning rate and sparsity ratio on fine-tuning models using LeZO on the SST-2 task. Results with accuracy exceeding 90% are displayed. Experiments were conducted with a single random seed. "Dropout Number" indicates the number of sparse layers in the OPT 13B model. To enhance clarity, the learning rates are logarithmically scaled. The curves mapping onto the straight lines in the lower plane are used to illustrate the range of learning rates that lead to improved performance after sparsifying different numbers of layers in the model. The performance of MeZO performance is the red line.

Figure 4: Correlation between the sparsity ratio and runtime in fine-tuning the OPT-13b model using LeZO on the SST-2 task. This figure presents the optimal experimental results at various sparsity levels from Figure 3, annotated with different colored pentagrams. The purple line indicates the total time required for fine-tuning in the corresponding experiments.

performance on the other datasets. Notably, LeZO and MeZO have the same memory consumption, and FT incurs a memory cost twelve times higher than LeZO.

**Performance on Multi-Size Models.** LeZO achieves a notable performance improvement over MeZO across models of varying scales. Tables 2 and 3 display the performance of LeZO and other methods on the OPT-1.3b and OPT-30b models, respectively. When fine-tuning OPT-1.3b on downstream tasks, LeZO achieves superior results in 9 out of 11 tasks. It achieves comparable results on the WSC task and its performance drop is less than 1% on RTE. When fine-tuning OPT-30b on the SST-2 and BoolQ datasets, LeZO demonstrates significant improvements compared to MeZO. The experiments across Tables 1, 2, and 3 reveal that LeZO achieves more pronounced performance gains on larger LLMs. This phenomenon may relate to the stability of model convergence, with larger models being more stable. Additionally, LeZO demonstrates greater stability compared to MeZO.

**Performance of Combining ZO optimizer and PEFT.** LeZO can be effectively integrated with PEFT methods. We combine ZO optimizers with PEFT by updating the trainable parameters introduced by PEFT across different methods. Table 4 showcases the fine-tuning results of LeZO and MeZO combined with the PEFT across different datasets. LeZO (LoRA) outperforms MeZO (LoRA) on all five datasets, while LeZO (prefix) outperforms MeZO (LoRA) on 4 out of 5 datasets. Additionally, the combination of LeZO with PEFT results in performance improvements exceeding 1% on the SST-2, CB, and BoolQ tasks compared to MeZO. This indicates that LeZO enhances fine-tuning effectiveness even when combined with PEFT.

**Convergence and Computation Speedup by LeZO.** LeZO is more efficient than MeZO. Fig-

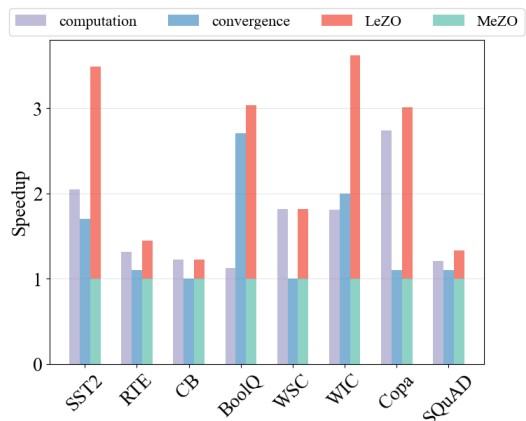

Figure 5: Comparison in computation efficiency between LeZO and MeZO on various tasks.

ure 1 illustrates the variation in accuracy on the test set when fine-tuning OPT-13b using LeZO on the SST-2 datasets. LeZO achieves $2\times$ computation speedup and $1.7\times$ convergence speedup. In

Figure 5, we present the convergence and computation speedups obtained when using LeZO to fine-tune OPT-13b on eight downstream tasks. The experimental results show that with the layer-wise sparsity policy, LeZO achieves effective convergence and computation speedups. In conclusion, LeZO achieves more efficient LLM fine-tuning than MeZO.

### 5.3 HYPERPARAMETER ANALYSIS

**Influence of Learning Rate and Sparsity Rate.** The layer-wise sparsity scheme we use shows how the number of sparse layers affects LeZO's performance. Our experiments show that with an increase in sparsity rate (more layers being sparsified), the ZO optimization process needs a higher learning rate. Figure 3 illustrates the relationship between the sparsity rate (Dropout Number), learning rate, and model accuracy on downstream tasks. From the figure, we obtain several key observations: **1)** LeZO demonstrates significantly superior optimization effectiveness compared to MeZO (Dropout Number=0). Under full-parameter fine-tuning, LeZO consistently outperforms MeZO regardless of the number of sparsified layers. **2)** LeZO enhances the robustness of the ZO optimization process. Comparing the lengths of the dashed lines, it can be observed that as the sparsity ratio increases, the range of learning rates that achieve over 90% accuracy on the SST-2 task expands. This indicates that the robustness of LeZO improves with an increased sparsity ratio. **3)** The behavior of LeZO markedly differs from fine-tuning only a subset of parameters. Performance experiences a substantial decline when the sparsity rate reaches $\rho = 1$ (sparse 40 layers in OPT-13b, while fine-tuning solely the embedding and linear layers).

**Impact of Downstream Tasks on Computational Speedup.** LeZO achieves varying speedup ratios across different tasks, as shown in Figure 5, which depicts the computation and convergence speedup ratios obtained by LeZO across all tasks. The differences in computational speedup ratios come from the varying proportions of the forward propagation process within the total computation. The speed of forward propagation depends on the average token length of different tasks within the same model framework. Figure 6 illustrates the relationship between the average input token length of different datasets and the computational speedup achieved by LeZO. When input token length increases, computation speedup decreases. With a constant sparsity ratio, LeZO's floating-point computational savings per downstream task remain fixed. Consequently, LeZO's computation acceleration is more suited for relatively straight-

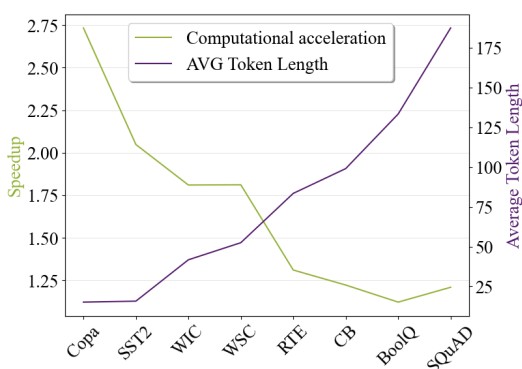

Figure 6: The relationship between the length of input tokens and the computational speedup achieved by LeZO.

forward training samples. However, a similar trend is not observed in the convergence acceleration of LeZO across different tasks. This disparity may be attributed to the varying difficulty levels of tasks, their convergence characteristics, and the distributions of training and extracted testing sets.

## 6 CONCLUSION

In this study, we introduce LeZO, a layer-wise sparse and efficient ZO optimizer for fine-tuning LLMs. We integrate the layer-wise pruning scheme into the ZO optimization process, which achieves both computation and convergence speedups in LLMs training without additional memory costs. Moreover, LeZO combines effectively with existing PEFT methods to further enhance acceleration. We conduct fine-tuning experiments on models from the OPT family using the Super-GLUE benchmark and two question-answering datasets. Empirical results show that LeZO significantly reduces training time and improves performance. Overall, LeZO provides an efficient and effective approach to ZO fine-tuning. It reduces the computational load caused by full-parameter perturbations in zeroth-order optimization without introducing additional memory overhead.

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

## A  DETAILED EXPERIMENTAL SETTINGS

We aligned the hyperparameter configurations for our primary experiments with the standards set by MeZO, as described by Malladi et al. (2023). Our experimental framework is based on the work of Zhang et al. (2024). Our approach incorporates a sparsity rate in LeZO. Our experiments indicate that this requires larger learning rates to enhance model convergence. Consequently, the grid search ranges for learning rates differ between LeZO and MeZO in the experiments presented in Table 1, 2, 3, and 4. We strictly adhered to the settings outlined by Malladi et al. (2023) when replicating MeZO experiments. The hyperparameter search ranges for grid search are detailed in Table 5. We conducted five trials with fixed random seeds for each set of experiments to compute average results and standard deviations. This facilitated a robust comparison of the efficiency of different methods. In the experiments depicted in Figure 3, due to the extensive number of trials, we conducted a single experiment with one random seed to obtain results under various hyperparameter settings. To streamline the evaluation process without compromising the representation of convergence across diverse models and datasets, we set the testing interval at 2000 steps. The convergence behavior of LeZO is contingent upon that of MeZO and is influenced by prompts; therefore, we did not conduct prompt removal experiments similar to those in MeZO. The prompts used in the experiments are identical to those in MeZO.

| Experiment | Hyperparameters | Values |
|---|---|---|
| LeZO | Batch size | 16 |
| | Learning rate | $\{1e-6, 7e-7\}$ for OPT-13b/30b, $\{3e-6, 1e-6\}$ for OPT-1.3b |
| | $\epsilon$ | $1e-3$ |
| | Sparse Rate | 0.75 |
| MeZO | Batch size | 16 |
| | Learning rate | $\{1e-6, 1e-7\}$ or $\{1e-6, 5e-7, 1e-7\}$ for RTE and SQuAD |
| | $\epsilon$ | $1e-3$ |
| LeZO (prefix) | Batch size | 16 |
| | Learning rate | $\{3e-2, 1e-2\}$ |
| | $\epsilon$ | $1e-1$ |
| | Sparse Rate | 0.75 |
| | # prefix tokens | 5 |
| MeZO (prefix) | Batch size | 16 |
| | Learning rate | $\{1e-2, 1e-3\}$ or $\{5e-2, 1e-2, 5e-3\}$ for SQuAD |
| | $\epsilon$ | $1e-1$ |
| | # prefix tokens | 5 |
| LeZO (LoRA) | Batch size | 16 |
| | Learning rate | $\{3e-5, 5e-5, 7e-5\}$ |
| | $\epsilon$ | $1e-2$ |
| | Sparse Rate | 0.50 |
| | $(r, \alpha)$ | $(8, 16)$ |
| MeZO (LoRA) | Batch size | 16 |
| | Learning rate | $\{1e-4, 5e-5\}$ or $\{1e-4, 5e-5, 1e-5\}$ for SQuAD |
| | $\epsilon$ | $1e-2$ |
| | $(r, \alpha)$ | $(8, 16)$ |
| FT with Adam | Batch size | 8 |
| | Learning Rate | $\{1e-5, 5e-5, 8e-5\}$ |

Table 5: The hyperparameter grids used for the experiments. Weight decay is set to 0. FT uses 5 epochs and linear scheduled learning rates. ZO optimizers use 20K steps and constant learning rates. We check validation performance and save the best checkpoint every 1/10 total training steps (2K).

## B  PROOF

From a single-step perspective, LeZO also involves updating a sub-network, which aligns with the theory of Sparse-MeZO (Liu et al., 2024). Therefore, the following proof draws inspiration from it.

*Proof of Lemma 1.* Let $\mathcal{L}_{\boldsymbol{z}}(\boldsymbol{\theta})$ be the expectation of $\mathcal{L}(\boldsymbol{\theta} + \epsilon \boldsymbol{z})$:

$$
\begin{aligned}
\mathcal{L}_{\mathcal{R}}(\boldsymbol{\theta}) :&= \mathbb{E}_z[\mathcal{L}(\boldsymbol{\theta} + \epsilon \mathcal{R}(\boldsymbol{z}))] \\
&= \mathbb{E}_{\mathcal{R}}[\mathcal{L}(\boldsymbol{\theta} + \epsilon \boldsymbol{z}^{'})].
\end{aligned}
\tag{10}
$$

Then,

$$
\begin{aligned}
\widehat{\nabla}_{\boldsymbol{\theta}'} \mathcal{L}_{\mathcal{R}}(\boldsymbol{\theta}) &= \widehat{\nabla}_{\boldsymbol{\theta}'} \mathbb{E}_{\mathcal{R}}[\mathcal{L}(\boldsymbol{\theta} + \epsilon \boldsymbol{z}^{'})] \\
&= \widehat{\nabla}_{\boldsymbol{\theta}'} \int_{\boldsymbol{z}'} \mathrm{pdf}_{\boldsymbol{z}'}(\boldsymbol{z}) \mathcal{L}(\boldsymbol{\theta} + \epsilon \boldsymbol{z}) d\boldsymbol{z} \\
&= \mathcal{R}(\nabla \int_{\boldsymbol{z}'} \mathrm{pdf}_{\boldsymbol{z}'}(z) \mathcal{L}(\boldsymbol{\theta} + \epsilon \boldsymbol{z}) dz) \\
&= \mathcal{R}(\int_{\boldsymbol{z}'} \nabla \mathrm{pdf}_{\boldsymbol{z}'}(z) \mathcal{L}(\boldsymbol{\theta} + \epsilon \boldsymbol{z}) dz) \\
&= \frac{1}{k} \mathcal{R}(\int_{\boldsymbol{z}'} \nabla e^{-\frac{1}{2}\|\boldsymbol{z}\|^2} \mathcal{L}(\boldsymbol{\theta} + \epsilon \boldsymbol{z}) dz) \\
&= \frac{1}{k} \mathcal{R}(\int_{\boldsymbol{y}'} \nabla e^{-\frac{1}{2}\|\frac{\boldsymbol{y}-\boldsymbol{\theta}}{\epsilon}\|^2} \mathcal{L}(\boldsymbol{y}) \frac{1}{\epsilon^n} d\boldsymbol{y}) \\
&= \frac{1}{k} \mathcal{R}(\int_{\boldsymbol{y}'} \frac{y - \boldsymbol{\theta}}{\epsilon^2} e^{-\frac{1}{2\epsilon^2}\|\boldsymbol{y}-\boldsymbol{\theta}\|^2} \mathcal{L}(\boldsymbol{y}) \frac{1}{\epsilon^n} d\boldsymbol{y}) \\
&= \frac{1}{k} \mathcal{R}(\int_{\boldsymbol{z}'} \frac{\boldsymbol{z}}{\epsilon} e^{-\frac{1}{2}\|\boldsymbol{z}\|^2} \mathcal{L}(\boldsymbol{\theta} + \epsilon \boldsymbol{z}) dz) \\
&= \mathcal{R}(\int_{\boldsymbol{z}'} \mathrm{pdf}_{\boldsymbol{z}'}(\boldsymbol{z}) \mathcal{L}(\boldsymbol{\theta} + \epsilon \boldsymbol{z}) \frac{\boldsymbol{z}}{\epsilon} dz) \\
&= \mathbb{E}_{\mathcal{R}}[\mathcal{R}(\frac{\mathcal{L}(\boldsymbol{\theta} + \epsilon \boldsymbol{z}^{'})}{\epsilon} \boldsymbol{z}^{'})] \\
&= \mathbb{E}_{\mathcal{R}}[\frac{\mathcal{L}(\boldsymbol{\theta} + \epsilon \boldsymbol{z}^{'})}{\epsilon} \boldsymbol{z}^{'}],
\end{aligned}
\tag{11}
$$

where $y = \boldsymbol{\theta} + \epsilon \boldsymbol{z}$, $\boldsymbol{y}^{'} = \boldsymbol{\theta} + \epsilon \boldsymbol{z}^{'}$, and $k = \sqrt{(2\pi)^{\rho d}}$.

Next,

$$
\begin{aligned}
\mathbb{E}_{\mathcal{R}}[\frac{\mathcal{L}(\boldsymbol{\theta} - \epsilon \boldsymbol{z}^{'})}{\epsilon} \boldsymbol{z}^{'}] &= \frac{1}{k} \int_{-\boldsymbol{z}'} \frac{\mathcal{L}(\boldsymbol{\theta} + \epsilon(-\boldsymbol{z}))}{\epsilon} - \boldsymbol{z} e^{-\frac{1}{2}\|-\boldsymbol{z}\|^2} d(-\boldsymbol{z}) \\
&= \frac{1}{k} \int_{\hat{\boldsymbol{z}}} \frac{\mathcal{L}(\boldsymbol{\theta} + \epsilon \boldsymbol{z})}{\epsilon} \boldsymbol{z} e^{-\frac{1}{2}\|z\|^2} dz \\
&= \mathbb{E}_{\mathcal{R}}[\frac{\mathcal{L}(\boldsymbol{\theta} + \epsilon \boldsymbol{z}^{'})}{\epsilon} \boldsymbol{z}^{'}].
\end{aligned}
\tag{12}
$$

Therefore,

$$
\begin{aligned}
\widehat{\nabla}_{\boldsymbol{\theta}'} \mathcal{L}_{\mathcal{R}}(\boldsymbol{\theta}) &= \mathbb{E}_{\mathcal{R}}[\frac{\mathcal{L}(\boldsymbol{\theta} + \epsilon \boldsymbol{z}^{'})}{\epsilon} \boldsymbol{z}^{'}] \\
&= \frac{1}{2}(\mathbb{E}_{\mathcal{R}}[\frac{\mathcal{L}(\boldsymbol{\theta} + \epsilon \boldsymbol{z}^{'})}{\epsilon} \boldsymbol{z}^{'}] - \mathbb{E}_{\mathcal{R}}[\frac{\mathcal{L}(\boldsymbol{\theta} - \epsilon \boldsymbol{z}^{'})}{\epsilon} \boldsymbol{z}^{'}]) \\
&= \mathbb{E}_{\mathcal{R}}[\frac{\mathcal{L}(\boldsymbol{\theta} + \epsilon \boldsymbol{z}^{'}) - \mathcal{L}(\boldsymbol{\theta} - \epsilon \boldsymbol{z}^{'})}{2\epsilon} \boldsymbol{z}^{'}] \\
&= \mathbb{E}_{\mathcal{R}}[\widehat{\nabla} \mathcal{L}_{\mathcal{R}}(\boldsymbol{\theta})].
\end{aligned}
\tag{13}
$$

Q.E.D. $\qquad\qquad\qquad\qquad\qquad\qquad\qquad\qquad\qquad\qquad\qquad\qquad\qquad\square$

*Proof of Lemma 2.* Firstly, compute the norm distance between the sparse ZO estimated gradient $\widehat{\nabla}_{\boldsymbol{\theta}'}\mathcal{L}_{\mathcal{R}}(\boldsymbol{\theta})$ and the sparse FO gradient $\widehat{\nabla}\mathcal{L}_{\mathcal{R}}(\boldsymbol{\theta})$:

$$
\begin{aligned}
\|\widehat{\nabla}_{\boldsymbol{\theta}'}\mathcal{L}_{\mathcal{R}}(\boldsymbol{\theta}) - \nabla\mathcal{L}_{\mathcal{R}}(\boldsymbol{\theta})\| &= \|\frac{1}{k}\int_{\boldsymbol{z}}(\frac{\mathcal{L}(\boldsymbol{\theta}+\epsilon\boldsymbol{z}) - \mathcal{L}(\boldsymbol{\theta}-\epsilon\boldsymbol{z})}{2\epsilon} - \langle\nabla\mathcal{L}_{\mathcal{R}}(\boldsymbol{\theta}), \boldsymbol{z}\rangle)\boldsymbol{z}e^{-\frac{1}{2}\|\boldsymbol{z}\|^2}d\boldsymbol{z}'\| \\
&= \|\frac{1}{k}\int_{\boldsymbol{z}}(\frac{\mathcal{L}(\boldsymbol{\theta}+\epsilon\boldsymbol{z}) - \mathcal{L}(\boldsymbol{\theta})}{\epsilon} - \langle\mathcal{R}(\nabla\mathcal{L}(\boldsymbol{\theta})), \boldsymbol{z}\rangle)\boldsymbol{z}e^{-\frac{1}{2}\|\boldsymbol{z}\|^2}d\boldsymbol{z}'\| \\
&\leq \frac{1}{k\epsilon}\int_{\boldsymbol{z}}|\mathcal{L}(\boldsymbol{\theta}+\epsilon\boldsymbol{z}) - \mathcal{L}(\boldsymbol{\theta}) - \epsilon\langle\nabla\mathcal{L}(\boldsymbol{\theta}), \epsilon\rangle|\|\mathcal{R}(\boldsymbol{z})\|e^{-\frac{1}{2}\|\boldsymbol{z}\|^2}d\boldsymbol{z}' \\
&\leq \frac{\epsilon L(l)}{2k}\int_{\epsilon}\|\boldsymbol{z}\|^2\|\mathcal{R}(\boldsymbol{z})\|e^{-\frac{1}{2}\|\boldsymbol{z}\|^2}d\boldsymbol{z}' \\
&= \frac{\epsilon L(l)}{2}\mathbb{E}_{\mathcal{R}}[\|\boldsymbol{z}'\|^3] \\
&\leq \frac{\epsilon L(l)}{2}(\rho d + 3)^{\frac{3}{2}}.
\end{aligned}
\tag{14}
$$

Subsequently, employing the Minkowski inequality for norms, which can be further generalized to $\|\boldsymbol{a}+\boldsymbol{b}\|^2 \leq 2\|\boldsymbol{a}\|^2 + 2\|\boldsymbol{b}\|^2$, by letting $\boldsymbol{a} = \nabla\mathcal{L}_{\mathcal{R}}(\boldsymbol{\theta}) - \widehat{\nabla}_{\boldsymbol{\theta}'}\mathcal{L}_{\mathcal{R}}(\boldsymbol{\theta})$ and $\boldsymbol{b} = \widehat{\nabla}_{\boldsymbol{\theta}'}\mathcal{L}_{\mathcal{R}}(\boldsymbol{\theta})$, we obtain:

$$
\begin{aligned}
\|\nabla\mathcal{L}_{\mathcal{R}}(\boldsymbol{\theta})\|^2 &\leq 2\|\nabla\mathcal{L}_{\mathcal{R}}(\boldsymbol{\theta}) - \widehat{\nabla}_{\boldsymbol{\theta}'}\mathcal{L}_{\mathcal{R}}(\boldsymbol{\theta})\|^2 + 2\|\widehat{\nabla}_{\boldsymbol{\theta}'}\mathcal{L}_{\mathcal{R}}(\boldsymbol{\theta})\|^2 \\
&= 2\|\widehat{\nabla}_{\boldsymbol{\theta}'}\mathcal{L}_{\mathcal{R}}(\boldsymbol{\theta}) - \nabla\mathcal{L}_{\mathcal{R}}(\boldsymbol{\theta})\|^2 + 2\|\widehat{\nabla}_{\boldsymbol{\theta}'}\mathcal{L}_{\mathcal{R}}(\boldsymbol{\theta})\|^2 \\
&\leq \frac{\epsilon^2 L^2(l)}{2}(\rho d + 3)^3 + 2\|\widehat{\nabla}\mathcal{L}_{\mathcal{R}}(\boldsymbol{\theta})\|^2 \\
&\leq \frac{\epsilon^2 L^2(l)}{2}(\rho d + 4)^3 + 2\|\widehat{\nabla}\mathcal{L}_{\mathcal{R}}(\boldsymbol{\theta})\|^2.
\end{aligned}
\tag{15}
$$

Q.E.D. $\qquad\square$

*Proof of Lemma 3.*

$$
\begin{aligned}
\mathcal{L}_{\mathcal{R}}(\boldsymbol{\theta}) - \mathcal{L}(\boldsymbol{\theta}) &= \mathbb{E}_{\mathcal{R}}[\mathcal{L}(\boldsymbol{\theta}+\epsilon\boldsymbol{z}') - \mathcal{L}(\boldsymbol{\theta})] \\
&= \mathbb{E}_{\mathcal{R}}[\mathcal{L}(\boldsymbol{\theta}+\epsilon\boldsymbol{z}') - \mathcal{L}(\boldsymbol{\theta}) - \epsilon\langle\nabla\mathcal{L}(\boldsymbol{\theta}), \boldsymbol{z}'\rangle] \\
&= \frac{1}{k}\int_{\boldsymbol{z}'}[\mathcal{L}(\boldsymbol{\theta}+\epsilon\boldsymbol{z}) - \mathcal{L}(\boldsymbol{\theta}) - \epsilon\langle\nabla\mathcal{L}(\boldsymbol{\theta}), \boldsymbol{z}\rangle]e^{-\frac{1}{2}\|\boldsymbol{z}\|^2}d\boldsymbol{z} \\
&\leq \frac{1}{k}\int_{\boldsymbol{z}'}\frac{\epsilon^2 L(l)}{2}\|\boldsymbol{z}\|^2 e^{-\frac{1}{2}\|\boldsymbol{z}\|^2}d\boldsymbol{z} \\
&= \frac{\epsilon^2 L(l)}{2}\mathbb{E}_{\mathcal{R}}[\|\boldsymbol{z}'\|^2] \\
&\leq \frac{\epsilon^2 L(l)}{2}\rho d,
\end{aligned}
\tag{16}
$$

where $\boldsymbol{\theta}_t = \boldsymbol{\theta} + \epsilon\boldsymbol{z}$. Given that $\mathcal{L}$ satisfies Lipschitz continuity, we can derive $|\mathcal{L}(\boldsymbol{\theta}') - \mathcal{L}(\boldsymbol{\theta}) - \langle\nabla\mathcal{L}(\boldsymbol{\theta}), \boldsymbol{\theta}_t - \boldsymbol{\theta}\rangle| \leq \frac{L(l)}{2}\|\boldsymbol{\theta}_t - \boldsymbol{\theta}\|^2$, establishing the validity of the first inequality.

$$
\begin{aligned}
[(\mathcal{L}_{\mathcal{R}}(\boldsymbol{\theta}) - \mathcal{L}(\boldsymbol{\theta})) - (\mathcal{L}_{\mathcal{R}}(\boldsymbol{\theta}+\epsilon\boldsymbol{z}') - \mathcal{L}(\boldsymbol{\theta}+\epsilon\boldsymbol{z}'))]^2 &\leq 2[\mathcal{L}_{\mathcal{R}}(\boldsymbol{\theta}) - \mathcal{L}(\boldsymbol{\theta})]^2 + 2[\mathcal{L}_{\mathcal{R}}(\boldsymbol{\theta}+\epsilon\boldsymbol{z}') - \mathcal{L}(\boldsymbol{\theta}+\epsilon\boldsymbol{z}')]^2 \\
&\leq \frac{\epsilon^4 L^2(l)}{2}\rho^2 d^2 + \frac{\epsilon^4 L^2(l)}{2}\rho^2 d^2 \\
&= \epsilon^4 L^2(l)\rho^2 d^2.
\end{aligned}
\tag{17}
$$

$$(\mathcal{L}_\mathcal{R}(\boldsymbol{\theta} + \epsilon \boldsymbol{z}') - \mathcal{L}_\mathcal{R}(\boldsymbol{\theta}))^2 \leq 2(\mathcal{L}_\mathcal{R}(\boldsymbol{\theta} + \epsilon \boldsymbol{z}') - \mathcal{L}_\mathcal{R}(\boldsymbol{\theta}) - \epsilon \langle \widehat{\nabla} \mathcal{L}_\mathcal{R}(\boldsymbol{\theta}), \boldsymbol{z}' \rangle)^2$$
$$+ 2(\epsilon \langle \widehat{\nabla} \mathcal{L}_\mathcal{R}(\boldsymbol{\theta}), \boldsymbol{z}' \rangle)^2$$
$$\leq \frac{\epsilon^4 L^2(l)}{2} \|\boldsymbol{z}'\|^4 + 2\epsilon^2 \langle \widehat{\nabla} \mathcal{L}_\mathcal{R}(\boldsymbol{\theta}), \boldsymbol{z}' \rangle^2 \tag{18}$$
$$\leq \frac{\epsilon^4 L^2(l)}{2} \|\boldsymbol{z}'\|^4 + 2\epsilon^2 \|\widehat{\nabla} \mathcal{L}_\mathcal{R}(\boldsymbol{\theta})\|^2 \|\boldsymbol{z}'\|^2.$$

$$(\mathcal{L}(\boldsymbol{\theta} + \epsilon \boldsymbol{z}') - \mathcal{L}(\boldsymbol{\theta}))^2$$
$$\leq 2((\mathcal{L}_\mathcal{R}(\boldsymbol{\theta}) - \mathcal{L}(\boldsymbol{\theta})) - (\mathcal{L}_\mathcal{R}(\boldsymbol{\theta} + \epsilon \boldsymbol{z}') - \mathcal{L}(\boldsymbol{\theta} + \epsilon \boldsymbol{z}')))^2$$
$$+ 2(\mathcal{L}_\mathcal{R}(\boldsymbol{\theta} + \epsilon \boldsymbol{z}') - \mathcal{L}_\mathcal{R}(\boldsymbol{\theta}))^2 \tag{19}$$
$$\leq 2\epsilon^4 L^2(l) \rho^2 d^2 + \epsilon^4 L^2(l) \|\boldsymbol{z}'\|^4 + 4\epsilon^2 \|\widehat{\nabla} \mathcal{L}_\mathcal{R}(\boldsymbol{\theta})\|^2 \|\boldsymbol{z}'\|^2$$

$$\mathbb{E}_{\boldsymbol{z},\boldsymbol{x}}[\|\widehat{\nabla} \mathcal{L}_\mathcal{R}(\boldsymbol{\theta})\|^2] = \mathbb{E}_\mathcal{R}[\|\frac{\mathcal{L}(\boldsymbol{\theta} + \epsilon \boldsymbol{z}') - \mathcal{L}(\boldsymbol{\theta} - \epsilon \boldsymbol{z}')}{2\epsilon} \boldsymbol{z}'\|^2]$$
$$= \mathbb{E}_\mathcal{R}[\|\frac{\mathcal{L}(\boldsymbol{\theta} + \epsilon \boldsymbol{z}') - \mathcal{L}(\boldsymbol{\theta})}{2\epsilon} \boldsymbol{z}' + \frac{\mathcal{L}(\boldsymbol{\theta}) - \mathcal{L}(\boldsymbol{\theta} - \epsilon \boldsymbol{z}')}{2\epsilon} \boldsymbol{z}'\|^2]$$
$$\leq \mathbb{E}_\mathcal{R}[2\|\frac{\mathcal{L}(\boldsymbol{\theta} + \epsilon \boldsymbol{z}') - \mathcal{L}(\boldsymbol{\theta})}{2\epsilon} \boldsymbol{z}'\|^2 + 2\|\frac{\mathcal{L}(\boldsymbol{\theta}) - \mathcal{L}(\boldsymbol{\theta} - \epsilon \boldsymbol{z}')}{2\epsilon} \boldsymbol{z}'\|^2]$$
$$= \mathbb{E}_\mathcal{R}[\frac{1}{2\epsilon^2}[\mathcal{L}(\boldsymbol{\theta} + \epsilon \boldsymbol{z}') - \mathcal{L}(\boldsymbol{\theta})]^2 \cdot \|\boldsymbol{z}'\|^2 + \frac{1}{2\epsilon^2}[\mathcal{L}(\boldsymbol{\theta}) - \mathcal{L}(\boldsymbol{\theta} - \epsilon \boldsymbol{z}')]^2 \cdot \|\boldsymbol{z}'\|^2]$$
$$\leq \mathbb{E}_\mathcal{R}[2\epsilon^2 L^2(l) \rho^2 d^2 \|\boldsymbol{z}'\|^2 + \epsilon^2 L^2(l) \|\boldsymbol{z}'\|^6 + 4\|\widehat{\nabla} \mathcal{L}_\mathcal{R}(\boldsymbol{\theta})\|^2 \|\boldsymbol{z}'\|^4]$$
$$\leq 2\epsilon^2 L^2(l) \rho^3 d^3 + \epsilon^2 L^2(l)(\rho d + 6)^3 + 4(\rho d + 4)^2 \|\widehat{\nabla} \mathcal{L}_\mathcal{R}(\boldsymbol{\theta})\|^2$$
$$\leq 3\epsilon^2 L^2(l)(\rho d + 4)^3 + 4(\rho d + 4)^2 \|\widehat{\nabla} \mathcal{L}_\mathcal{R}(\boldsymbol{\theta})\|^2. \tag{20}$$

As $\mathbb{E}_\mathcal{R}[\|\boldsymbol{z}'\|^p] \leq (\rho d + p)^{\frac{p}{2}}$ for $p \geq 2$, the third inequality holds. Additionally, since $2\rho^3 d^3 + (\rho d + 6)^3 \leq 3(\rho d + 4)^3$, the fourth inequality is valid.

Again, given that $\mathcal{L}$ is Lipschitz continuous, we have $|\mathcal{L}(\boldsymbol{\theta}_{t+1}) - \mathcal{L}(\boldsymbol{\theta}_t) - \langle \nabla \mathcal{L}(\boldsymbol{\theta}_t), \boldsymbol{\theta}_{t+1} - \boldsymbol{\theta}_t \rangle| \leq \frac{L(l)}{2} \|\boldsymbol{\theta}_{t+1} - \boldsymbol{\theta}_t\|^2$. Therefore, we can derive:

$$\mathcal{L}_\mathcal{R}(\boldsymbol{\theta}_{t+1}) - \mathcal{L}_\mathcal{R}(\boldsymbol{\theta}_t) - \langle \widehat{\nabla} \mathcal{L}_\mathcal{R}(\boldsymbol{\theta}_t), \boldsymbol{\theta}_{t+1} - \boldsymbol{\theta}_t \rangle$$
$$\leq |\mathcal{L}_\mathcal{R}(\boldsymbol{\theta}_{t+1}) - \mathcal{L}_\mathcal{R}(\boldsymbol{\theta}_t) - \langle \widehat{\nabla} \mathcal{L}_\mathcal{R}(\boldsymbol{\theta}_t), \boldsymbol{\theta}_{t+1} - \boldsymbol{\theta}_t \rangle| \tag{21}$$
$$\leq \frac{L(l)}{2} \|\boldsymbol{\theta}_{t+1} - \boldsymbol{\theta}_t\|^2.$$

By further examining the iterative process of Structured Sparse ZO-SGD as outlined in Equation (5), we can obtain:

$$\mathcal{L}_\mathcal{R}(\boldsymbol{\theta}_{t+1}) \leq \mathcal{L}_\mathcal{R}(\boldsymbol{\theta}_t) + \langle \widehat{\nabla} \mathcal{L}_\mathcal{R}(\boldsymbol{\theta}_t), \boldsymbol{\theta}_{t+1} - \boldsymbol{\theta}_t \rangle + \frac{L(l)}{2} \|\boldsymbol{\theta}_t - \boldsymbol{\theta}_{t+1}\|^2$$
$$= \mathcal{L}_\mathcal{R}(\boldsymbol{\theta}_t) - \eta_t \langle \widehat{\nabla} \mathcal{L}_\mathcal{R}(\boldsymbol{\theta}_t), \widehat{\nabla} \mathcal{L}_\mathcal{R}(\boldsymbol{\theta}_t) \rangle + \frac{(\eta_t)^2 L(l)}{2} \|\widehat{\nabla} \mathcal{L}_\mathcal{R}(\boldsymbol{\theta}_t)\|^2, \tag{22}$$

where $\eta_t$ represents the learning rate at step $t$.

Subsequently, we can derive the expected loss function of the structured sparse model at step $t + 1$ as:

$$\mathbb{E}_{\boldsymbol{z}',\boldsymbol{x}}[\mathcal{L}_{\mathcal{R}}(\boldsymbol{\theta}_{t+1})] \leq \mathbb{E}_{\boldsymbol{z}',\boldsymbol{x}}[\mathcal{L}_{\mathcal{R}}(\boldsymbol{\theta}_t)] - \eta_t \mathbb{E}_{\boldsymbol{z}',\boldsymbol{x}}[\|\widehat{\nabla}\mathcal{L}_{\mathcal{R}}(\boldsymbol{\theta}_t)\|^2]$$

$$+ \frac{(\eta_t)^2 L(l_{\boldsymbol{z}})}{2} \mathbb{E}_{\boldsymbol{z}',\boldsymbol{x}}[\|\widehat{\nabla}\mathcal{L}(\boldsymbol{\theta}_t)\|^2]$$

$$\leq \mathbb{E}_{\boldsymbol{z}',\boldsymbol{x}}[\mathcal{L}_{\mathcal{R}}(\boldsymbol{\theta}_t)] - \eta_t \mathbb{E}_{\boldsymbol{z}',x}[\|\widehat{\nabla}\mathcal{L}_{\mathcal{R}}(\boldsymbol{\theta}_t)\|^2] \tag{23}$$

$$+ \frac{(\eta_t)^2 L(l)}{2}(4(\rho d + 4)\mathbb{E}_{\boldsymbol{z}',x}[\|\widehat{\nabla}\mathcal{L}_{\mathcal{R}}(\boldsymbol{\theta}_t)\|^2] + 3\epsilon^2 L^2(l)(\rho d + 4)^3).$$

Then, let learning rate be $\eta_t = \frac{1}{4(\rho d + 4)L(l)}$ and obtain:

$$\mathbb{E}_{\boldsymbol{z}',\boldsymbol{x}}[\mathcal{L}_{\mathcal{R}}(\boldsymbol{\theta}_{t+1})] \leq \mathbb{E}_{\boldsymbol{z}',\boldsymbol{x}}[\mathcal{L}_{\mathcal{R}}(\boldsymbol{\theta}_t)] - \frac{1}{8(\rho d + 4)L(l)}\mathbb{E}_{\boldsymbol{z}',\boldsymbol{x}}[\|\widehat{\nabla}\mathcal{L}_{\mathcal{R}}(\boldsymbol{\theta}_t)\|^2] + \frac{3\epsilon^2}{32}L(l)(\rho d + 4). \tag{24}$$

Summing Equation (24) from 0 to $T + 1$, where $T$ denotes a sufficiently large number of training steps, yields:

$$\mathbb{E}_{\boldsymbol{z}',\boldsymbol{x}}[\|\widehat{\nabla}\mathcal{L}_{\mathcal{R}}(\boldsymbol{\theta}_T)\|^2] \leq 8(\rho d + 4)L[\frac{\mathcal{L}_{\mathcal{R}}(\boldsymbol{\theta}_0) - \mathcal{L}_{\mathcal{R}}^*}{T + 1} + \frac{3\epsilon^2}{32}L(\rho d + 4)], \tag{25}$$

where $L(l) \leq L$ for all $\mathcal{L}(\boldsymbol{\theta}_t)$. Thus, based on Lemma 2, we can have:

$$\mathbb{E}_{\boldsymbol{z}',\boldsymbol{x}}[\|\nabla\mathcal{L}_m(\boldsymbol{\theta}_T)\|^2] \leq \frac{\epsilon^2 L^2}{2}(\rho d + 4)^3 + 2\mathbb{E}_{\boldsymbol{z}',\boldsymbol{x}}[\|\widehat{\nabla}\mathcal{L}_{\mathcal{R}}(\boldsymbol{\theta}_T)\|^2]$$

$$\leq 16(\rho d + 4)L\frac{\mathcal{L}_{\mathcal{R}}(\boldsymbol{\theta}_0) - \mathcal{L}_{\mathcal{R}}^*}{T + 1} + \frac{\epsilon^2 L^2}{2}(\rho d + 4)^2(\rho d + \frac{11}{2}). \tag{26}$$

To obtain $\sigma$-accurate solution $\mathbb{E}_{\boldsymbol{z}',x}[\|\nabla\mathcal{L}_m(\boldsymbol{\theta}_T)\|^2] \leq \sigma^2$, we can define $\epsilon = \Omega(\frac{\sigma}{\rho^{\frac{3}{2}}d^{\frac{3}{2}}L})$.

$$16(\rho d + 4)L\frac{\mathcal{L}_{\mathcal{R}}(\boldsymbol{\theta}_0) - \mathcal{L}_{\mathcal{R}}^*}{T + 1} + \mathcal{O}(\epsilon^2 L^2 \rho^3 d^3)$$

$$= 16(\rho d + 4)L\frac{\mathcal{L}_{\mathcal{R}}(\boldsymbol{\theta}_0 - \mathcal{L}_{\mathcal{R}}^*)}{T + 1} + \mathcal{O}(\sigma^2). \tag{27}$$

From the above, we can get:

$$T = \mathcal{O}(\frac{\rho d L}{\sigma^2}). \tag{28}$$

Q.E.D. $\qquad\qquad\qquad\qquad\qquad\qquad\qquad\qquad\qquad\qquad\qquad\qquad\qquad\qquad\qquad\square$

