# OpenReview forum: "Simultaneous Computation and Memory Efficient Zeroth-Order Optimizer for Fine-Tuning Large Language Models"
_ICLR.cc/2025/Conference — ICLR 2025 Conference Withdrawn Submission_

### Official Review · Reviewer_3vj3 · 2024-10-23

**Soundness:** 2
**Presentation:** 3
**Contribution:** 2
**Rating:** 3
**Confidence:** 4

**Summary:**

The paper aims to reduce the computational cost of zeroth-order (ZO) optimization, specifically building upon the memory-efficient ZO (MeZO) optimizer. The main idea of the paper involves incorporating a layer-wise sparsity by randomly selecting a subset of layers to be tuned at each fine-tuning step. Experiments show that the proposed method, LeZO, achieves a noticeable speedup in fine-tuning LLMs while retaining comparable performance.

**Strengths:**

S1: Paper is overall well written and organized, providing some nice background and discussion of related works.

S2: Extensive experiments on different model size scales, discussion on evaluating LeZO on orthogonal PEFT methods, and analysis of the impacts of hyperparameters.

**Weaknesses:**

W1: Contribution lacks some novelty as components are borrowed from closely related work. For instance, LISA proposed by Pan et. al. 2024, uses a similar layer-wise sampling that is applied to first order optimizers like AdamW. The convergence analysis presented in the paper is also based on Sparse-MeZO by Liu et al. 2024, with minor changes.

W2: Although Sparse-MeZO was mentioned as a related work, it was not used as a baseline to evaluate against. Including this can be important to observe performance gaps between the two approaches, which both apply a form of parameter selection to improve MeZO.

**Questions:**

Q1: There seems to be a typo in Table 1. The approach "SSZO" is used but not referenced or explained in any other part of the paper. If this is a typo meant to refer to LeZO, then the authors should double-check for consistency in naming.

Q2: In Figure 4, the drop in accuracy between dropout number 0 and 40 is only less than 0.5%, which does not seem significant considering all the layers have been frozen in fine-tuning. Can this be attributed to the fact that you are still fine-tuning the embedding and linear layers as mentioned in section 5.3? How much of an impact are the results obtained impacted by this?

---

### Official Review · Reviewer_bAuv · 2024-10-31

**Soundness:** 2
**Presentation:** 2
**Contribution:** 2
**Rating:** 3
**Confidence:** 4

**Summary:**

The paper addresses the challenge of high memory usage during the fine-tuning of large language models (LLMs). It revisits the Memory-efficient Zeroth-Order (MeZO) optimizer and identifies that the perturbation and updating processes consume over 50% of the fine-tuning time.
To mitigate this, the authors propose LeZO, a layer-wise sparse computation and memory-efficient zeroth-order optimizer. LeZO introduces dynamic layer-wise sparsification, treating layers as fundamental units and perturbing different parameter subsets in each step to achieve full-parameter fine-tuning.
The proposed method aims to reduce computational costs without additional memory overhead and demonstrates faster convergence than MeZO on the SuperGLUE benchmark and two generative tasks.

**Strengths:**

- The authors conduct extensive experiments on the OPT model family using the SuperGLUE benchmark and two generative tasks. The reported 3× speedup over MeZO on tasks like SST-2, BoolQ, and Copa provides strong empirical evidence for the effectiveness of LeZO.

- LeZO effectively integrates layer-wise sparsity into the simultaneous perturbation stochastic approximation (SPSA) and zeroth-order stochastic gradient descent (ZO-SGD), maintaining the theoretical foundations of ZO optimization.

**Weaknesses:**

- The theory is based on the assumption of Lipchitz continuity. In the context of deep learning and LLMs, this assumption can be overly simplistic or unrealistic. Loss landscapes in deep neural networks are often non-convex and may not satisfy Lipschitz continuity, not to say the billion params LLMs.

- The paper primarily compares LeZO with MeZO. Including comparisons with other state-of-the-art optimization techniques for fine-tuning LLMs, such as first-order methods (e.g., Adam, AdamW) or other memory-efficient optimizers, would provide a more comprehensive evaluation. (the LLM community are still generally using AdamW, I am not convinced if this appeals to adam users)

- The claim that LeZO achieves accelerated computation without additional memory overhead could be elaborated upon.

- The setting of LoRA could sway the performances by a lot with different rank and alpha, the paper only tested using r=8, and alpha=16, I would suggest try different LoRA set ups and offer evidence that it could work in the higher rank setting. Along the same vain, as the eval is done mainly by evaluating downstream performance, then adding more models into comparison would be more convincing.

**Questions:**

- how much /where Lezo saved memories on? any detailed profiling?

---

### Official Review · Reviewer_m7AX · 2024-11-02

**Soundness:** 2
**Presentation:** 2
**Contribution:** 3
**Rating:** 5
**Confidence:** 3

**Summary:**

This paper proposes a novel zeroth-order optimizer, named LeZO, which enhances the efficiency of the Memory-efficient Zeroth-Order (MeZO) optimizer by introducing sparsity in the parameter updates. The key innovation of LeZO is that it applies sparsification at the layer level, treating each layer as a unit of sparse updating. Through analysis, the authors observe that more than half of MeZO’s computational cost is spent on perturbation and updating stages. By selectively updating only a subset of layers in each iteration, LeZO significantly reduces this overhead, while still ensuring that all parameters are eventually updated. This strategy avoids additional memory costs, such as those incurred by masking. Experimental results demonstrate that LeZO achieves up to a 3.4× speedup in training wall-clock time compared to the original MeZO, without compromising optimization performance.

**Strengths:**

1. The proposed method is straightforward and practical, making it easy to integrate into existing training frameworks with minimal engineering effort. By sparsifying updates at the layer level, it achieves efficiency without complex changes, making it highly usable.
2. Although writing clarity could be improved, the paper is organized logically, which helps convey the main ideas and findings in a generally understandable way.
3. Experimental results show clear benefits: the approach reduces computational overhead in perturbation and updating, achieving up to 3.4× faster wall-clock training times, highlighting its effectiveness in accelerating convergence.

**Weaknesses:**

1. Inconsistent Terminology and Notation: The paper suffers from inconsistent use of terms and symbols. For instance, the term "sparse rate $\rho$" on Line 194 and "sparsity rate reaches $\rho = 1$" on Line 451 seem to refer to entirely different concepts, which may confuse readers.
2. Errors and Ambiguities in Mathematical Notation: The mathematical expressions and symbols lack rigor and contain errors. According to Line 195, $\theta' \in \mathbb{R}^{\rho d}$ while $\theta \in \mathbb{R}^{d}$, making the addition of $\theta$ and $z'$ in Equation (4) undefined, as these vectors lie in different spaces. Additionally, cases where $\rho d$ is not an integer are not addressed. Notation in Lemma 1 is also quite unclear, making it difficult to interpret.
3. Errors in Tables: Table 1 contains unexplained elements, such as the appearance of "SSZO," which is not introduced or defined, potentially causing confusion.

**Questions:**

1. In Line 310, it is stated that as the sparsity ratio $\rho$ decreases, the upper bound on the convergence time to the expected value also decreases. If $\rho$ here is defined as on Line 194, then when $\rho = 0$, meaning no parameters are perturbed, why would the upper bound on the required convergence time be minimized in this case?
2. Your method appears quite similar to sparse MeZO, with convergence analysis and proofs that are almost identical. Sparse MeZO selects parameters for updates based on their magnitudes, while your approach randomly selects layers for updates. Why didn’t you include a comparison with sparse MeZO? Does your method offer any performance advantages, or is the primary benefit a reduction in memory overhead? Additionally, is there any theoretical insight for choosing layers as the basic unit for sparsification?
3. In Figure 3, there are notable performance drops between dropout numbers of 5 and 0 (MeZO), suggesting a significant performance gain by excluding updates for just 5 layers. There are also marked drops between dropout numbers 35 and 40. How do you explain these observations?

---

### Official Review · Reviewer_aNSB · 2024-11-04

**Soundness:** 3
**Presentation:** 3
**Contribution:** 2
**Rating:** 5
**Confidence:** 3

**Summary:**

The authors propose LeZO, a method that integrates the ideas of BCD and ZO-SGD to accelerate training time in comparison to MeZO by Malladi et al. (2023).

**Strengths:**

1. The algorithm is straightforward and easy to understand, combining BCD with MeZO.
2. The authors provide a theoretical analysis of the convergence rate.
3. They empirically demonstrate that their method achieves a speed-up in fine-tuning experiments on the OPT model family.

**Weaknesses:**

1. The method consistently improves the convergence rate compared to MeZO; however, the rate still scales with d without further assumptions.
2. Building on point 1, the improvement in convergence rate is clear both theoretically and empirically. However, the corresponding impact on model performance remains unclear and unexplained.
3. There is a lack of empirical results on larger models, such as 30B. Additionally, testing on model types other than OPT should be considered.
4. Is the random selection of parameters optimal? Alternative selection methods, such as using importance sampling and weight norms, remain unexplored.

**Questions:**

See the above weaknesses for questions.

---

### Note · Authors · 2024-11-14

I have read and agree with the venue's withdrawal policy on behalf of myself and my co-authors.